# Novel Resting Energy Expenditure Prediction Equations for Multi-Ethnic Asian Older Adults with Multimorbidity

**DOI:** 10.3390/nu17132144

**Published:** 2025-06-27

**Authors:** Pei San Kua, Musfirah Albakri, Su Mei Tay, Phoebe Si-En Thong, Olivia Jiawen Xia, Wendelynn Hui Ping Chua, Kevin Chong, Nicholas Wei Kiat Tan, Xin Hui Loh, Jia Hui Tan, Lian Leng Low

**Affiliations:** 1SingHealth Community Hospitals, 10 Hospital Blvd, Singapore 168582, Singapore; musfirah.albakri@singhealthch.com.sg (M.A.); tay.su.mei@singhealthch.com.sg (S.M.T.); phoebe.thong.s.e@singhealthch.com.sg (P.S.-E.T.); olivia.xia.j.w@singhealth.com.sg (O.J.X.); wendelynn.chua.h.p@singhealth.com.sg (W.H.P.C.); kevin.chong@singhealthch.com.sg (K.C.); nicholas.tan.w.k@singhealthch.com.sg (N.W.K.T.); loh.xin.hui@singhealthch.com.sg (X.H.L.); tan.jia.hui@singhealthch.com.sg (J.H.T.); low.lian.leng@singhealth.com.sg (L.L.L.); 2SingHealth Centre for Population Health Research and Implementation, 10 Hospital Boulevard, #19-01 SingHealth Tower, Singapore 168582, Singapore

**Keywords:** resting energy expenditure, prediction equations, Asian older adults, multimorbidity, mid-upper arm circumference, calf circumference

## Abstract

Background/Objectives: Malnutrition is prevalent among hospitalized older adults with multimorbidity, leading to adverse health outcomes and increased healthcare costs. An accurate assessment of resting energy expenditure (REE) is crucial because an inaccurate estimation of energy needs may result in unintentional underfeeding or overfeeding, both of which can worsen nutritional status and clinical outcomes. While indirect calorimetry (IC) is the preferred method, its clinical applicability is limited. Prediction equations are commonly used, but their accuracy in older Asian patients with multimorbidity remains unclear. Methods: This multicenter, cross-sectional study enrolled 400 patients aged ≥65 years from March to December 2023 in Outram Community Hospital (OCH) and Sengkang Community Hospital (SKCH). Participants’ REE was measured using indirect calorimetry. We compared the performance of the newly developed novel prediction equations (PEs), derived from readily accessible or easily measured anthropometric data, against established equations. Statistical analysis included the calculation of R^2^, the root mean square error (RMSE), and the intraclass correlation coefficient (ICC) to assess reliability and goodness of fit. Results: A high prevalence (85%) of multimorbidity was observed among the participants. REE increased progressively with body mass index (BMI) across all groups (865.6–1269.4 kcal in females; 889.1–1269.4 kcal in males). The novel PEs (RMSE: 186–191; ICC: 0.5–0.52) demonstrated improved accuracy and stronger reliability compared to conventional equations (RMSE: 222–258; ICC: 0.271–0.460). Conclusions: Our newly developed PEs offer potentially valuable tools for precise REE estimation in hospitalized older Asian patients with multimorbidity. Further external validation and investigation in diverse populations are necessary to confirm these results.

## 1. Introduction

Multimorbidity can be defined as at least two co-occurring chronic diseases present in the same person [1]. Hospitalized older patients, who often present with a variety of chronic conditions, have higher risks of malnutrition. This can lead to increased healthcare burden, with increased mortality and morbidity rates, hospital length of stay, re-hospitalization, and hospital expenditure [2]. It has been observed that early nutritional support in multimorbid patients can reduce the risk of sarcopenia and adverse outcomes by providing at least 75% of energy requirements [1]. While energy requirements decline with age, it is still crucial to prevent malnutrition and maintain body reserves through a valid estimation of energy requirements [3].

Resting energy expenditure (REE) is defined as the energy required to sustain and maintain the integrity of vital functions when an individual is completely at rest [4]. REE is the largest component of an individual’s daily energy expenditure, at 60–75% [5]. Additionally, fat-free mass (FFM) decreases with age and may affect REE due to its reduced quantity and metabolic activity [3,5]. Indirect calorimetry is the gold standard for measuring REE; however, it is often impractical in a clinical setting, and considerable expertise in required to execute it [1,4,6]. Hence, in hospitals, dietitians often rely on established prediction equations, which include covariates such as age, sex, weight and height multiplied by activity, and stress factors to estimate an individual’s energy requirements. Equations often used in Singapore healthcare institutions include the Harris–Benedict [7], Schofield [8], and Mifflin–St Jeor [9] equations. Another option is weight-based [1] formulas for 18–20 kcal/kg body weight, while 30 kcal/kg body weight can be used in severely underweight patients with multimorbidity [1]. However, commonly used prediction equations, such as Harris–Benedict [7], Schofield [8], and Mifflin–St Jeor [9], were developed for generally healthy people and are not exclusive to Asians or older adults [7,8,9]. Numerous prediction equations published in the literature show an accuracy (predicted values falling within 90% to 110% of the measured REE, with values outside this range considered inaccurate) of 36–75% in multimorbid patients, and this accuracy is worse compared to that of indirect calorimetry [1]. There is also a lack of clarity on which prediction equation is the most reliable for older adults [3]. While there is an equation that was previously developed for use in Singapore, it was developed for the Chinese population aged 21–67 years; therefore, it may not be the most appropriate one to be used for older patients and those of other ethnicities [10].

Energy prediction equations are important in developing a nutrition care plan, particularly in the geriatric population, where malnutrition and frailty risk are high [11]. Appropriate nutrition interventions should be promptly put in place to prevent further healthcare burden. With the worldwide increase in the aging population, the development of specialized PEs for this growing segment of the population has become increasingly critical [12]. To our knowledge, our work is the first to evaluate the accuracy of PEs for estimating REE in hospitalized, multi-ethnic Asian older adults with multimorbidity. Our study also aims to contribute to the limited body of evidence by developing and validating a context-specific equation for use in rehabilitation settings.

## 2. Materials and Methods

### 2.1. Study Design

This was a multicenter, cross-sectional study involving patients admitted for rehabilitative and subacute care at SKCH and OCH in Singapore between March and December 2023. Participants were eligible if they were aged 65 years or older, medically stable as assessed by physicians, free of multidrug-resistant organisms (MDROs), not terminally ill, able to eat orally, and capable of providing informed consent. Patients were excluded if they were on enteral feeding, medically unstable (e.g., febrile, desaturated, or experiencing any symptoms screened by doctors that deemed them unfit for indirect calorimetry measurement), terminally ill with a life expectancy of less than six months, or if they declined or withdrew consent. Written informed consent was obtained from all participants. For patients lacking mental capacity, consent was obtained from their next of kin or legal guardian. Protocol of this study was approved by the SingHealth Centralised Institutional Review Board (CIRB).

### 2.2. Resting Energy Expenditure Measurement

Prior to the REE measurement, participants were required to fast for 6–8 h and were advised to avoid moderate to vigorous physical activity one day prior and to rest 10 to 15 min prior to the measurement. REE measurement was conducted using a Q-NRG^TM^ (COSMED Ltd., Rome, Italy) indirect calorimetry machine in an isolated, quiet area with an ambient temperature of 23–25 °C from 7:00 h to 8:00 h. Participants were in supine position for 20 min while being covered in a canopy (as per the machine instructions) after 5 min of stabilization under standardized conditions. A trained investigator was present to ensure the participant was awake throughout the measurement. REE was calculated from the volume of oxygen consumption (VO_2_) and volume of carbon dioxide produced (VCO_2_) using Weir’s formula [13]. Monthly calibration of flow and gas analyzers was carried out based on manufacturer’s instructions.

### 2.3. Anthropometry Measurements, Nutrition Assessment Tool, and Study Variables

Prior to but within the same week of REE measurement, body weight was measured in light hospital clothes to the nearest 0.1 kg using electronic scale chair scales (SECA 952, SECA GmbH & Co. KG, Hamburg (district Eilbek), Germany) for participants who were able to ambulate out of bed or using a patient lifter (Paramount Bed KQ-781, Paramount Bed Co., Ltd., Tokyo, Japan) for those who were bed-bound and advised for non-weight bearing during their hospital stay. Height was measured without shoes to the nearest 0.1 cm using measuring tape.

On the same day as REE measurements, participants were asked to sit on their beds or on the geriatric chairs at their bedsides to measure their mid-upper arm circumference (MUAC) and calf circumference (CC) using an anthropometry tape to the nearest 0.1 cm. MUAC was measured at the mid-point between the acromial surface of the scapula and the olecranon process of the elbow over the dominant arm when bent at 90 degrees, while CC was measured using non-elastic tape at the widest part of the dominant calf in the seated position with the weight evenly distributed on both feet [14]. A 7-point Subjective Global Assessment (SGA) was conducted by trained dietitians to assess the nutrition status of participants [15] within the same week of REE measurement. Participants’ clinical characteristics were extracted from the Electronic Health Intelligence System (eHINTS) with the approval of the ethics board.

### 2.4. Statistical Analysis

As the older adult population of Singapore is large and increasing, the Cochran formula [*n* = (*Z*_0.95_)^2^
*P* [(1 − *P*)/*D*^2^] is appropriate [16]. By using estimates by Chan et al. [17], which indicated that 39% of older patients in step-down and long-term care are undernourished, 366 patients were the minimum number of patients required for this study. Our recruited sample size was 400 participants, well above this threshold, providing a robust sample size.

Continuous variables were summarized using mean and standard deviation or median and interquartile range, as appropriate. Categorical variables were presented as frequencies and percentages. To investigate the relationship between various factors and REE, two statistical approaches were utilized. To explore the different utilities of parameters obtained from physical examinations and electronic medical records, we constructed two distinct regression models with varying predictor sets. First, a multivariate generalized linear model was implemented. Second, a multivariate regression model with restricted cubic splines was constructed to account for potential non-linear relationships between continuous predictors and REE.

Initial candidate models included demographic and anthropometric variables that are commonly collected in PEs: age, sex, weight, and height. To explore the impact of additional clinical factors, variables which were used to assess the participants’ muscle mass and nutritional status were also considered, specifically CC, MUAC, and SGA. Diabetes mellitus and cancer, prevalent chronic diseases associated with significant metabolic derangements and alterations in body composition, were also included in this analysis [4,18]. Ordinary squares linear regression was performed first, followed by backward elimination, with a liberal removal criterion of *p* > 0.2, which was used to identify the most parsimonious model. This process resulted in four multivariable final models, with two PEs including age, weight, and height and two other PEs including age, sex, MUAC, and CC. A variance inflation factor (VIF) analysis was conducted to assess collinearity in each set, and the values ranged from 1.03 to 4.64, indicating low collinearity. Additionally, an ANOVA test was conducted within each set to check on the linearity of the variables in order to determine the suitability of splining the relevant variables.

All models were validated using internal bootstrap validation with 1000 resampling iterations to assess their performance [19]. The choice of iterations was selected to enhance the stability and precision of the estimated performance metrics. This was followed by the validation of the predicted REE obtained from the Harris–Benedict [7], Schofield [8], Mifflin–St Joer [9], and weight-based methods [1] for participants and from the newly developed models using the measured REE as reference. Individual prediction accuracy was defined as the percentage of subjects that had a predicted REE within ±10% of the measured REE, and prediction below 90% of measured REE was classified as under-prediction, while a prediction above 110% was deemed as over-prediction [3,10].

The coefficients of determination (r^2^), the intraclass correlation coefficient (ICC) between REE values measured by IC and REE values calculated using various Pes, and the root mean square error (RMSE) were used to define the predictions obtained from these models. The guideline for interpreting ICC suggests that a value of 0.81–1.00 indicates almost perfect agreement, 0.61–0.80 indicates substantial agreement, 0.41–0.60 indicates moderate agreement, 0.21–0.40 indicates fair agreement, and a value of less than 0.21 is regarded as poor agreement [20]. Significant two-tailed *p* values were set at <0.05. In addition, Bland–Altman analysis was performed to evaluate the agreement between measured REE (REE-IC) and predicted REE values derived from both conventional and newly developed predictive equations. R studio (RStudio 2023.12.1 Build 402 “Ocean Storm” for Windows) was used for all analyses.

## 3. Results

As shown in Figure 1, a total of 2278 patients were admitted to OCH and SKCH for rehabilitation and subacute care from March to December 2023. A total of 816 participants were deemed eligible based on the inclusion criteria. Of these, 45 patients were withdrawn due to reasons such as inability to adhere to fasting requirements, testing positive for MDROs after recruitment, feeling anxious during measurement, and discomfort from gastrointestinal issues. In total, 400 participants completed the measurements. However, three participants were eventually excluded due to missing indirect calorimetry and calf circumference (CC) data.

In our study of 397 participants, the Chinese and Malay groups had more females (Chinese: [63%] vs. males: 133 [37%]; Malay: 12 [63%] vs. 7 [37%]), while the Indian group had slightly more males (11 [52%] vs. females: 10 [48%]). The Eurasian group had equal sex representation (one each). Table 1 shows that males were generally heavier (median 61 kg vs. 53 kg) and taller (median 165 cm vs. 152 cm), resulting in a lower median BMI (22 vs. 24 kg/m^2^). Both sexes had similar median SGA scores (6, indicating potential malnutrition risk) [14]. Females showed a higher median Modified Barthel Index, MBI (64 vs. 59), suggesting greater independence in daily living [21]. Males had a slightly larger median CC (33 vs. 32 cm), while MUAC was the same (27 cm). Median CC and MUAC measurements were near or below sarcopenia risk cutoffs for both sexes, indicating potential increased risk [22,23].

Further analysis revealed a high prevalence of multimorbidity within the study population. A total of 85% of participants presented with at least two co-existing chronic conditions. Hypertension emerged as the most prevalent comorbidity, affecting 73.0% of participants. Hyperlipidemia (62.2%) and diabetes mellitus (35.8%) were also frequently observed. Other prevalent chronic conditions included osteoarthritis (35.5%), osteoporosis (25.7%), chronic kidney disease (26.7%), cancer (16.9%), and gout (8.6%).

Statistical analysis (Table 2) showed that REE increased across BMI for both female and males. Age, weight, height, SGA, MUAC, and calf circumference significantly impacted measured REE (Table 3). However, sex, race, hypertension, hyperlipidemia, diabetes, and cancer did not significantly affect REE.

To account for potential challenges in acquiring routine weight and height measurements in some clinical settings, we developed four REE prediction models based on dietitian recommendations and commonly collected variables. Two models utilized age, weight, and height, while the remaining two models included age, sex, MUAC, and CC. The models (Model 1 and 2) were initially presented in their generalized linear equation forms (PE 1 and PE 2, respectively), without the inclusion of spline transformation.
New Prediction Equation 1 (Model 1):REE (kcal/day) = 2420.11 + 34.13 (W) − 5.65 (A) − 18.31 (H),
New Prediction Equation 2 (Model 2):REE (kcal/day) = 812.67 + 12.61 (M) + 13.65 (C) − 57.5 (S) − 6.99 (A),
where W = weight (kg); A = age (years); H = height (H); S = sex (male = 0; female = 1); M = MUAC (cm); C = CC (cm).

Preliminary regressions using four- and five-knot splines were conducted in addition to the three-knot model to determine the optimal knot placement. The results indicated that increasing the number of knots did not yield substantial improvements in model performance and even introduced a higher risk of overfitting. Based on the calibration curves plotted, the three-knot spline was therefore selected as it offered the optimal balance between model accuracy and complexity. For easier utility, we generated PE 1 and PE 2 in the form of nomograms (with splines) as Model 3 and Model 4 to estimate REE probability from age, sex, weight, height, MUAC, and CC [24]. Point values assigned to each predictor were summed to yield a total score, which was then converted to a predicted probability or outcome using a reference scale (Figure 2).

Given the complexity and potential for error in manual REE calculations, we developed a web-based application using Shiny in R to streamline this process for PE 1 and PE 2 (with splines) as Model 5 and Model 6 [25]. This tool provides a user-friendly interface for data entry, automates the calculation based on established equations, and delivers immediate results. This approach promotes accuracy, saves time, and enhances the accessibility of REE determination for both research and clinical settings. The link to PE 1 (Model 5) is PE 1 and PE 2 was used for PE 2 (Model 6). These models were then validated and their performance metrics are presented in Table 4. This table provides a comparative analysis of the models’ accuracy in estimating REE in a cohort of 397 participants, relative to established PEs. The table presents the mean predicted REE and standard deviation for each equation, along with the percentage of accurate predictions (within 10% of measured REE), under- and over-prediction percentages, R-squared (R^2^) values, root mean squared error (RMSE), and intraclass correlation coefficients (ICCs) with 95% confidence intervals.

The new PEs demonstrate a substantial gain in accuracy (46–48% accurate predictions) compared to traditional equations: Schofield [8] (35%), Mifflin–St Jeor [9] (38%), weight-based [1] (39%), and Harris–Benedict [7] (41%). A comparison of RMSE and ICC was conducted to evaluate the accuracy of prediction equations. The new PEs presented lower RMSE values (186–191) than other methods (222–258), indicating better accuracy. The ICC, which reflects reliability, showed moderate agreement (0.5–0.52) for the new equations, comparable to some existing methods and better than others, suggesting that the new equations are more reliable and accurate for REE prediction.

Furthermore, Harris–Benedict [7], Schofield [8], Mifflin–St Jeor [9], and weight-based [1] PEs exhibited notable mean biases (ranging from −152.9 to −18.1 kcal) and wide 95% limits of agreement, indicating substantial variability at the individual level (Figure 3a). In contrast, the new PE 1 and 2 demonstrated negligible mean bias (−0.6 to 0.0 kcal) and narrower limits of agreement (362–368 kcal), suggesting improved agreement with measured REE (Figure 3b). PE 1 and 2 (both direct equations and nomograms) showed the tightest limits of agreement and minimal proportional bias. These findings support the utility of the new equations, especially PE 1, as more accurate and reliable alternatives to existing PEs.

## 4. Discussion

### 4.1. Rationale for Developing New REE PEs

Our study introduces new predictive equations (PE 1 and PE 2) for REE, available in direct and nomogram or webpage formats for an improved nutritional assessment in Asian multiethnic older adults undergoing rehabilitation. It is important to address this population because the accuracy of predicting REE in Asian older adults using equations developed from European/North American populations (Harris–Benedict [7], Schofield [8], Mifflin–St Jeor [9]) is a concern. Our study revealed a tendency for these equations to overestimate REE in older adults, consistent with prior research suggesting a potential overestimation of basal metabolic rate (BMR) in Asian populations when using the commonly applied Harris–Benedict predictive equation [26]. Furthermore, although the Schofield database included thousands of participants, 45% were of Italian origin, a group reported to have significantly higher BMR compared to other ethnicities, and this likely contributed to the Schofield equation overestimating BMR in our Asian older adult population [27]. In contrast to a previous study [28], our study found that the weight-based [4] PE over-predicts REE in heavier older adults. This formula’s reliance on total body weight, without accounting for the relative proportions of muscle and fat, likely results in REE overestimation in older adults. This population typically experiences a shift towards a higher fat-to-muscle ratio, which the formula does not adequately address [29].

### 4.2. Performance of New PEs

Our results shows that the new PEs correct the over-prediction tendency of traditional methods, showing a more balanced distribution of under- and over-predictions. By reducing this bias, our PEs offer a valuable tool for preventing overfeeding and subsequent weight gain, especially in overweight patients with hip or knee surgeries or injuries with limited mobility, in whom weight management is crucial [30,31]. PE 1 demonstrated improved accuracy for REE estimation compared to standard equations. Given the common availability of age, weight, and height data in polyclinics and hospitals, PE 1 is readily adaptable for implementation within electronic medical record systems, alongside established equations such as Harris–Benedict [7], Mifflin–St Jeor [8], or Schofield [9]. PE 2, which omits weight and height, showed comparable accuracy to PE 1. This makes PE 2 advantageous when weight and height measurements are impractical. Specifically, PE 2 avoids weight requirements, a common limitation in older adult assessments with limited outpatient access.

Our new PE 2 utilizes MUAC to replace weight in contrast to a prior equation that utilized both variables in Japanese older adults. This simplifies assessments and may enhance muscle mass evaluation and reduce the impact of fluid retention [30]. It must be acknowledged that Kawase et al.’s PE exhibited superior accuracy in their study population [28]. However, our PE 2 has a similar RMSE when compared with another PE by Priscila et al., which also uses MUAC as part of their PE [4]. While PE 1 and PE 2 were slightly better (r^2^ = 0.34) than existing PEs, their overall predictive power was still low. This suggests that predicting REE is complex and requires considering many factors, including body composition, individual activity levels, and hormonal influences, which were not included in our equations [32,33,34].

### 4.3. Easy Application of New PEs

We have developed new PEs based on common anthropometry data to facilitate their practical application in healthcare institutions and within the broader community. This approach ensures ease of implementation and encourages widespread use in clinical practice. PE 1 is specifically designed for settings such as hospitals and clinics, where age, weight, and height data are routinely collected. Recognizing that accurate weight measurements can be challenging in certain situations, PE 2 incorporates calf circumference and mid-upper arm circumference as alternative input parameters. These measurements are non-invasive, easily administered by healthcare professionals or caregivers, and contribute to improved usability. To maximize accessibility, we have also developed nomograms and web-based platforms to enable calculations for users who may not have access to complete clinical documentation.

### 4.4. Factors Influencing REE Prediction Models

The univariate analysis identified age, sex, weight, height, calf circumference, and mid-upper arm circumference as factors influencing REE. The well-documented decline in REE with age is largely explained by changes in fat-free mass (FFM) [30,35]. Weight is associated with REE due to its composition of muscle, FFM, and fat mass, which contribute to heat production [36]. Height’s influence on REE stems from the larger organ and body mass in taller individuals [36]. Additionally, calf and mid-upper arm circumferences, indicators of metabolically active muscle mass, reinforce the relationship between muscle mass and REE [37,38]. Therefore, our new PEs incorporate these significant variables: age, weight, height, MUAC, and CC.

### 4.5. Strengths, Limitations, and Future Research

Our study addresses a gap in current knowledge by investigating the association between readily obtainable clinical determinants and objectively measured REE in a population of older Asian adults with multimorbidity, a highly relevant demographic in our healthcare system. Our research offers valuable insights for healthcare professionals to estimate energy expenditure in the Asian older adults with multimorbidity, enabling them to create more individualized nutrition plans for older adults in various care settings, complementing existing nutritional guidelines. The comparable accuracy of PE 1 and PE 2 allows for their interchangeable application, ensuring seamless continuous care for individual patients from the hospitals, clinics, nursing homes, and even home-based care.

We also acknowledge limitations that may affect the generalizability of its findings. This study employed a single IC measurement due to the inherent challenges associated with prolonged fasting protocols in older adults. This approach limits our ability to fully capture the variability in REE. Additionally, the lack of body composition data, ideally obtained through dual-energy X-ray absorptiometry (DXA), hinders a more comprehensive understanding of REE [39]. Furthermore, the limited representation of severely malnourished and underweight individuals highlights the need for further research focused on populations with lower BMI or confirmed malnutrition, as their REE may differ significantly [40].

Furthermore, the Bland–Altman plots reveal a proportional bias in the newly developed PEs, with REE underestimated at lower metabolic rates and overestimated at higher rates. Ideally, differences should be evenly distributed around the zero line; however, the observed trend suggests a reduced accuracy at the extremes of REE, like in the previous study [41]. Future studies should consider refining the equations using a more diverse samples, particularly including individuals with very low or high metabolic rates to minimize this bias and enhance prediction accuracy across the full REE spectrum. Lastly, the study recognizes REE as a component of total daily energy expenditure (TDEE) [42]. Future investigations should prioritize developing more accurate methods to assess physical activity levels and stress factors in older patients, alongside implementation studies on the feasibility, adoption, and effectiveness of REE equations. These factors are crucial for dietitians to precisely calculate individual TDEE and optimize nutritional interventions.

## 5. Conclusions

In conclusion, our novel prediction equations offer a significant advancement in REE estimation for older adults with multimorbidity, demonstrating superior reliability and accuracy compared to established formulas. The incorporation of easily accessible anthropometric data, particularly calf and mid-upper arm circumference, enhances usability across diverse settings, especially when traditional measurements are challenging. While these findings are promising, further validation in long-term care settings is crucial to confirm their broad applicability for community-dwelling older adults.

## Figures and Tables

**Figure 1 nutrients-17-02144-f001:**
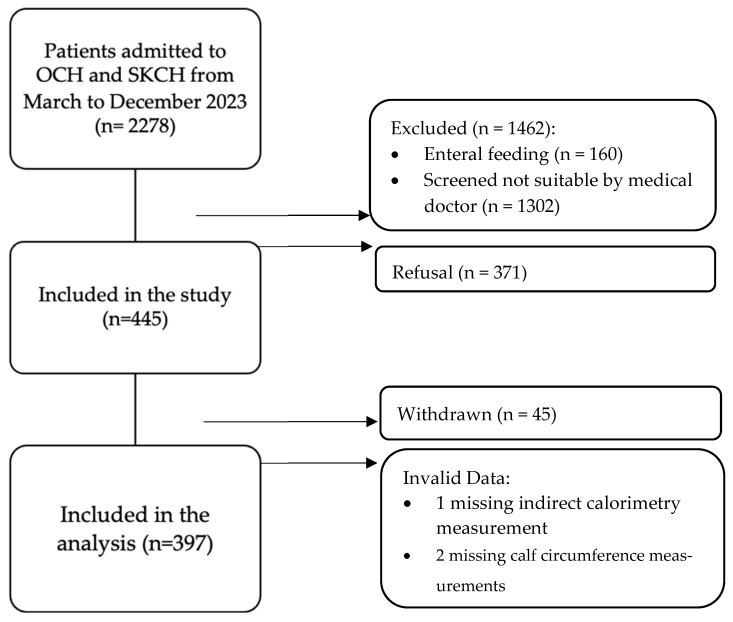
Patient selection flowchart.

**Figure 2 nutrients-17-02144-f002:**
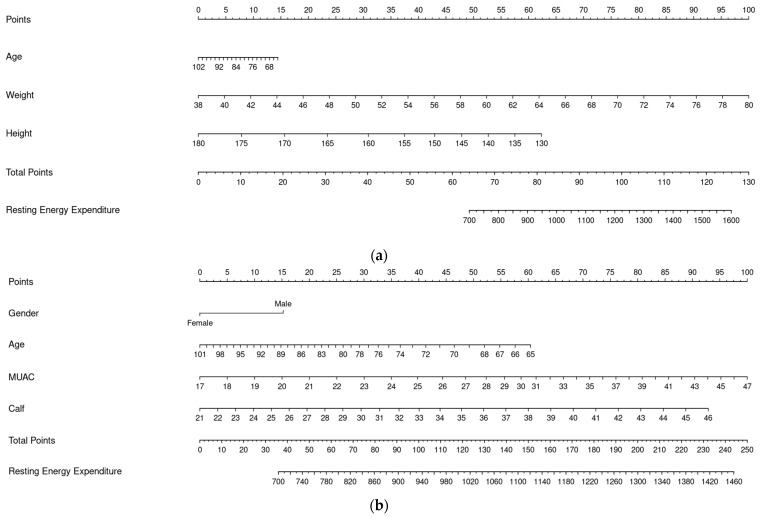
Nomograms for PEs (Model 3 and Model 4). (**a**) Nomogram for PE 1 (Model 3). (**b**) Nomogram for PE 2 (Model 4).

**Figure 3 nutrients-17-02144-f003:**
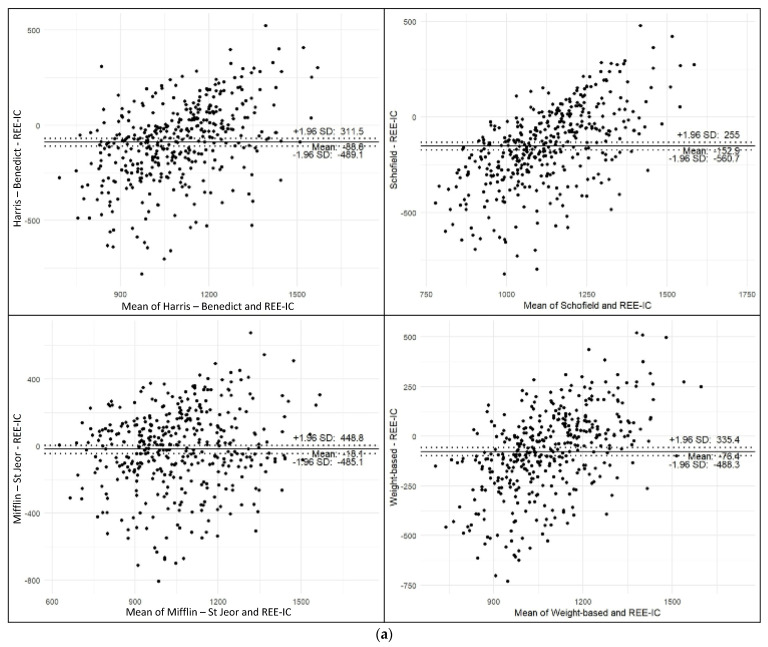
Bland–Altman Analysis. (**a**) Bland–Altman plots comparing predicted and measured REE using conventional equations. (**b**) Bland–Altman plots comparing predicted and measured REE using newly developed PEs.

**Table 1 nutrients-17-02144-t001:** Patient characteristics.

	Total (n = 397)				
	Male		Female		
	Median	IQR	Median		IQR
Age (year)	75	71–82	73		69–79
Weight (kg)	61	57–65	53		50–57
Height (cm)	165	160–169	152		148–156
Body Mass Index (kg/m^2^)	22	20–26	24		21–28
7-point Subjective Global Assessment (SGA)	6	5–7	6		6–7
Modified Barthel Index (MBI)	59	47–73	64		46–76
Calf circumference (CC, cm)	33	30–35	32		30–35
Mid-upper Arm Circumference (MUAC, cm)	27	25–30	27		25–31

**Table 2 nutrients-17-02144-t002:** Mean resting energy expenditure (REE) by BMI category and stratified by sex.

BMI Category	Male (n = 152)		Female (n = 245)		Total (n = 397)	
	Mean REE (kcal)	±SD	Mean REE (kcal)	±SD	Mean REE (kcal)	±SD
Underweight	889.1	214.2	865.6	141.0	874.1	169.2
Normal	1009.0	217.8	933.8	158.3	968.4	191.0
Overweight	1136.0	191.2	1042.8	186.4	1077.8	192.9
Obese	1269.4	276.7	1219.1	207.0	1233.4	228.4

**Table 3 nutrients-17-02144-t003:** Regression analysis of factors associated with energy requirement.

Variable	Estimate (95% CI)	*p*-Value
Age	−11.05 (−14.22, −7.89)	<0.005
Weight	14.60 (11.59, 17.60)	<0.005
Height	5.11 (2.60, 7.61)	<0.005
Subjective Global Assessment (SGA)	93.81 (67.48, 120.15)	<0.005
Mid-upper Arm Circumference (MUAC)	24.82 (20.51, 29.12)	<0.005
Calf Circumference (CC)	26.89 (22.26, 31.51)	<0.005
Sex (Female)	−44.15 (−90.59, 2.28)	0.06
Presence of Hypertension	−3.42 (−54.51, 47.67)	0.895
Presence of Hyperlipidemia	−24.06 (−70.76, 22.64)	0.311
Presence of Diabetes mellitus	−9.153 (−56.44, 38.14)	0.704
Presence of Cancer	−37.01 (−97.43, 23.41)	0.229

**Table 4 nutrients-17-02144-t004:** Performance of new PEs for REE (n = 397).

	Equations	REE (kcal)	Accurate Prediction (%) ^i^	Under-Prediction (%) ^ii^	Over-Prediction (%) ^iii^	R-Squared	RMSE (kcal) ^iv^	ICC [IC 95%] ^v^
**Total**	REE-IC	1050.4 ± 229.2						
**(n = 397)**	Harris–Benedict [7]	1139.2 ± 149.4	43%	14%	44%	0.06	222	0.403 ♣ [0.26–0.519]
	Schofield [8]	1203.3 ± 124	35%	18%	57%	−0.08	258	0.271 ◆ [0.045–0.449]
	Mifflin–St Jeor [9]	1068.6 ± 200.6	38%	31%	31%	−0.27	239	0.389 ◆ [0.302–0.469]
	Weight-based [1]	1125.6 ± 203.3	39%	19%	42%	0.05	232	0.460 ♣ [0.349–0.554]
	PE 1 (equation; Model 1)	1050.9 ± 137.4	48%	24%	28%	0.31	186	0.529 ♣ [0.454–0.597]
	PE 1 (nomogram, web; Model 3, 5)	1050.9 ± 136	48%	24%	27%	0.31	187	0.522 ♣ [0.446–0.59]
	PE 2 (equation; Model 2)	1050.4 ± 132.1	48%	25%	28%	0.34	191	0.5 ♣ [0.422–0.57]
	PE 2 (nomogram, web; Model 4, 6)	1050.4 ± 131.6	47%	26%	27%	0.34	190	0.497 ♣ [0.419–0.567]

^i^ Percentage of participants predicted by this predictive equation within 10% of the measured value. ^ii^ Percentage of participants predicted by this predictive equation within <10% of the measured value. ^iii^ Percentage of participants predicted by this predictive equation within >10% of the measured value. ^iv^ Root mean squared prediction error (RMSE). ^v^ Intraclass correlation coefficient (ICC, with 95% confidence interval): ♣ 0.41–0.60: Moderate agreement. ◆ 0.21–0.40: Fair agreement.

## Data Availability

The raw data supporting the conclusions of this article will be made available by the authors on request.

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
