# Peer review of "Novel Resting Energy Expenditure Prediction Equations for Multi-Ethnic Asian Older Adults with Multimorbidity"

_nutrients, 2025, doi:10.3390/nu17132144_

Round 1
Reviewer 1 Report
Comments and Suggestions for Authors
This paper presents a clinically relevant issue—malnutrition in elderly multimorbid patients and the need for accurate REE prediction.
The initial sentence of the abstract could be more concise. Consider stating the problem and its impact more directly.
Suggested revision: “Malnutrition is common in hospitalized elderly patients with multimorbidity, leading to poor outcomes and increased healthcare costs. The transition from the problem (malnutrition) to the focus on REE prediction methods is somewhat abrupt. Better linkage could strengthen the rationale. Clarify that malnutrition management depends on accurately estimating energy needs, hence the importance of REE.
A study cannot be both cross-sectional (single time point) and prospective (followed over time). Please clarify whether this was a single-time-point assessment across multiple centers.
The result stating "average REE was 1050.4 ± 229.2 kcal/day" is helpful, but it's unclear how this varies by sex, BMI category, or comorbidity burden.
The phrase “offer potentially valuable tools” is appropriate, but you need to temper claims such as “contribute to optimized nutritional support” unless outcome benefits were actually measured (which seems unlikely in a cross-sectional design).
Clarify study design and patient selection criteria.
Provide methodological transparency on how the new equations were derived and validated.
For greater insight, consider including subgroup analysis (e.g., by sex, BMI, and disease burden) in the final manuscript.
Author Response
Comment 1: The initial sentence of the abstract could be more concise. Consider stating the problem and its impact more directly.
Response 1: Thank you for pointing this out, we agree with this comment. Therefore, we have changed the initial sentence in Page 1, line 11-13 to “Malnutrition is prevalent among hospitalized older adults with multimorbidity, leading to adverse health outcomes and increased healthcare costs.”
Comment 2: The transition from the problem (malnutrition) to the focus on REE prediction methods is somewhat abrupt. Better linkage could strengthen the rationale. Clarify that malnutrition management depends on accurately estimating energy needs, hence the importance of REE.
Response 2: Agree, we have added “Accurate assessment of resting energy expenditure (REE) is crucial because inaccurate estimation of energy needs may result in unintentional underfeeding or overfeeding, both of which can worsen nutritional status and clinical outcomes.” to the Page 1, line 13-15.
Comment 3: A study cannot be both cross-sectional (single time point) and prospective (followed over time). Please clarify whether this was a single-time-point assessment across multiple centers.
Response 3: Thank you for spotting this. As this is a single point, cross sectional study, we have removed the word “prospective” in Page 1, both lines 18 and 94.
Comment 4: The result stating "average REE was 1050.4 ± 229.2 kcal/day" is helpful, but it's unclear how this varies by sex, BMI category, or comorbidity burden.
Response 4: Agree. We have included “REE increased progressively with body mass index (BMI) across all groups (865.6–1269.4 kcal in females; 889.1–1269.4 kcal in males).” to the Page 1, line 26-27. We also added similar statement in Page 6, line 255-256 and added the summary of this analysis in Page 6, line 260, Table 2: Mean Resting Energy Expenditure (REE) by BMI Categories and Stratified by Sex.
Comment 5: The phrase “offer potentially valuable tools” is appropriate, but you need to temper claims such as “contribute to optimized nutritional support” unless outcome benefits were actually measured (which seems unlikely in a cross-sectional design).
Response 5: Thank you for the suggestion. We changed the sentence in Page 1 from 32 to 33 to “Our newly developed PEs offer potentially valuable tools for precise REE estimation in hospitalized older Asian patients with multimorbidity.”.
Comment 6: Clarify study design and patient selection criteria.
Response 6: We have revised the Study Design section in Page 2 - 3, lines 94 - 104 in accordance with your feedback.
Comment 7: Provide methodological transparency on how the new equations were derived and validated. For greater insight, consider including subgroup analysis (e.g., by sex, BMI, and disease burden) in the final manuscript.
Response 7: Thank you for your comment. We have revised the sentences in Page 3 to 4, lines 147-148, 150-153, 158–166, 168-169, 183-185, Page 6 to 7, line 264–288, 291-292, 301-303 and 306–310. These edits were made to improve clarity, particularly regarding the methodology. Additionally, as suggested, we have included Table 2 (in Page 6, line 260) to present the subgroup analysis results.
Additional Clarififcation
In addition to the amendments mentioned above, we also revised the citation sequence due to the removal of four references from the abstract. This adjustment affected citations 1–9, which have been corrected on Pages 1–2 and 13.
Reviewer 2 Report
Comments and Suggestions for Authors
This study develops prediction equations for estimating resting metabolic rate in hospitalized Asian older adults.
The manuscript is generally well written, but I suggest including Bland- Altman plots to compare predicted resting metabolic rate and measured resting metabolic rate differences.
If possible, include the two novel prediction equations in the abstract
Line 46: change “ have an effect” to “affect”
Line 50 should gender be changed to sex?
Line 58. How are you defining accuracy here?
Line 93 subscript the 2 in VO2 and CO2
Line 94 here it is stated the calibration was done monthly. Should this be done each day?
Line 103 clarify how you identified the site for upper arm circumference and calf circumferences
Liner 126 and 133 should gender be changed to sex here? Sex is biological. Gender is a behavioural characteristic.
Line 130 I suggest deleting the word “therefore”
Line 159 change “dat” to data
Figure 1 legend: define CC in the legend
Lines176, 179, 183, 192, 199 and elsewhere: again I suggest changing gender to sex
Line 180 make sure to define the abbreviation MBI
In a footnote to table 2, define all abbreviations used in the table
Line 212 should CC23 be changed to CC?
Footnote to table 3: not all of the symbols in the footnote are used in the table, and not all of the symbols used in the table are defined in the footnote
Line 304: change “ to prior an equation” to “ to a prior equation”
Line 347 make sure that you have defined the abbreviation IF in the manuscript
Author Response
Comment 1: The manuscript is generally well written, but I suggest including Bland- Altman plots to compare predicted resting metabolic rate and measured resting metabolic rate differences.
Response 1: Thank you for the feedback. We have included the Bland-Altman (BA) analysis in the methodology section (Page 4, Lines 182–184), and added the corresponding BA plots in the results section (Pages 9–10, Lines 349–355).
Comment 2: If possible, include the two novel prediction equations in the abstract
Response 2: Thank you very much for your valuable suggestion. We explored the possibility of including all six models in the abstract; however, this would have exceeded the journal’s word limit of 250 words. Therefore, we regretfully decided not to proceed with this inclusion.
Comment 3: Line 46: change “ have an effect” to “affect”
Response 3: Amended Page 2 (previous line 46), line 53 to “affect”.
Comment 4: Line 50 should gender be changed to sex?
Response 4: Agree, replaced “gender” with “sex” on Page 2 (previous line 50, current line 57).
Comment 5: Line 58. How are you defining accuracy here?
Response 5: As noted in the original citation, the authors assessed the accuracy of various prediction equations by comparing predicted and measured REE, emphasizing the variability in predictive performance—defined as values within 90% to 110% of the measured REE—and the extent of under- or overestimation across patient groups. Predictions outside this range were considered inaccurate. Accordingly, we have revised the text on Page 2 (previously line 58; currently lines 64–68) to accurately reflect this.
Comment 6: Line 93 subscript the 2 in VO2 and CO2
Response 6: Sure, we have revised Page 3—previously Line 93, now Line 115—in accordance with your feedback.
Comment 7: Line 94 here it is stated the calibration was done monthly. Should this be done each day?
Response 7: Thank you for double-checking this detail. The QNRG+ requires only monthly manual calibration of the flowmeter and gas analyzer. This model features automated daily calibration, eliminating the need for manual user intervention (source: COSMED Q-NRG Technical Specifications).
Comment 8: Line 103 clarify how you identified the site for upper arm circumference and calf circumferences
Response 8: Sure, we have updated the sentence and citation on Page 3, Lines 127–130 to specify the measurement sites for mid-upper arm and calf circumference.
Comment 9: Liner 126 and 133 should gender be changed to sex here? Sex is biological. Gender is a behavioural characteristic.
Response 9: Yes, we have replaced all instances of 'gender' with 'sex' throughout the manuscript.
Comment 10: Line 130 I suggest deleting the word “therefore”
Response 10: Yes, we have removed the word 'therefore' from Page 4—previously Line 130, now Line 157.
Comment 11: Line 159 change “dat” to data
Response 11: Yes, we have corrected to “data” from Page 4-previous line 159, current line 193.
Comment 12: Figure 1 legend: define CC in the legend
Response 12: Sure, we have included “calf circumference measurements” in Figure 1.
Comment 13: Lines176, 179, 183, 192, 199 and elsewhere: again I suggest changing gender to sex
Response 13: Yes, we have replaced all instances of 'gender' with 'sex' throughout the manuscript.
Comment 14: Line 180 make sure to define the abbreviation MBI
Response 14: Yes, we have included “Modified Barthel Index” prior mentioning MBI in the paragraph from Page -previous line 180, current line 247.
Comment 15: In a footnote to table 2, define all abbreviations used in the table
Response 15: Agree, updated footnote to Table 2.
Comment 16: Line 212 should CC23 be changed to CC?
Response 16: Thank you for highlighting this. We have corrected “CC23” to “CC [23]” on Page 7, line 288 (previously Page 5, line 212), as “23” refers to the citation.
Comment 17: Footnote to table 3: not all of the symbols in the footnote are used in the table, and not all of the symbols used in the table are defined in the footnote
Response 17: We appreciate your observation and have made the necessary corrections to the symbols.
Comment 18: Line 304: change “ to prior an equation” to “ to a prior equation”
Response 18: We have accurately revised the sentence at previous line 304, now located on Page 11, line 392.
Comment 19: Line 347 make sure that you have defined the abbreviation IF in the manuscript
Response 19: Yes, we have defined the abbreviation 'IC' as indirect calorimetry in the abstract on Page 1, line 16.
Additional Clarifiations
In addition to the amendments mentioned above, we also revised the citation sequence due to the removal of four references from the abstract. This adjustment affected citations 1–9, which have been corrected on Pages 1–2 and 13.
Reviewer 3 Report
Comments and Suggestions for Authors
Dear Authors,
The article addresses a current and relevant issue from the perspective of dietetics, clinical nutrition, and public health. The methodology used is generally correct, and the presentation of the results is clear, although your manuscript needs some clarification.
Please note:
1) Avoiding the phrase “elderly,” as it is pejorative and leads to discrimination against older people. Instead, I suggest using words such as: older patient, senior, older adult, etc.
2) Please complete the affiliations, that is, add the full addresses of the institutions you represent.
3) I usually avoid citations in the abstract. I would suggest providing RMSE and ICC values and not pointing to specific equations with citations. This can be done later in the article.
4) In the introduction, please clearly indicate the study's originality.
5) Please clarify the sampling criteria in the materials, methods, and exclusion criteria section.
6) Please improve the resolution of Figure 2.
7) I suggest enriching the discussion by updating the literature review in the discussion section with the latest publications of max 5 years.
Best regards,
The reviewer.
Author Response
Comment 1: Avoiding the phrase “elderly,” as it is pejorative and leads to discrimination against older people. Instead, I suggest using words such as: older patient, senior, older adult, etc.
Response 1: Thank you for pointing this out. We have replaced the term “elderly” with “older patient” or “older adult” in the following locations: Page 1, lines 11, 17, 33, and 42; Page 2, lines 69 and 71; Page 3, line 138; and Page 12, lines 429, 437, and 448.
Comment 2: Please complete the affiliations, that is, add the full addresses of the institutions you represent.
Response 2: The affiliation has been updated in Page 1, line 6 in accordance with your feedback.
Comment 3: I usually avoid citations in the abstract. I would suggest providing RMSE and ICC values and not pointing to specific equations with citations. This can be done later in the article.
Response 3: We have removed the citations in Page 1, line 22 and 30 based on your suggestion.
Comment 4: In the introduction, please clearly indicate the study's originality.
Response 4: Agree, we have added “To our knowledge, our work is the first study to evaluate the accuracy of PEs for estimating REE in hospitalized older multi-ethnic Asian adults with multimorbidity. Our study also aims to contribute to the limited body of evidence by developing and validating a context-specific equation for use in rehabilitation settings.” to Page 2, line 78-81.
Comment 5: Please clarify the sampling criteria in the materials, methods, and exclusion criteria section.
Response 5: The Study Design section (Page 2 to 3, lines 94–104) has been revised in response to your feedback.
Comment 6: Please improve the resolution of Figure 2.
Response 6: Thank you for pointing this out. We have replaced Figure 2 (Page 7, line 299) with a higher-resolution image.
Comment 7: I suggest enriching the discussion by updating the literature review in the discussion section with the latest publications of max 5 years.
Response 7: We have revised the content and citations on Page 10, lines 366-372, and on Page 11, lines 417–423. Additionally, we have updated the citations on Page 11-12, lines 419-447. The modified references include citation 27 (Page 14, lines 563-564) and citations 36–41 (Page 14-15, lines 586–601).
Additional Clarifications
In addition to the amendments mentioned above, we also revised the citation sequence due to the removal of four references from the abstract. This adjustment affected citations 1–9, which have been corrected on Pages 1–2 and 13.
Round 2
Reviewer 2 Report
Comments and Suggestions for Authors
Thanks for responding to my previous comments. I have a few additional suggestions for revision:
Line 64: insert a space between the words “patients” and “and”.
Line 76: Delete the “g” at the start of this line.
Line 96: For VO2 and VCO2 you have superscripted the “2”. This should be subscripted instead. Please correct this throughout.
Figure 1: This flow chart now runs off the page and is mis-aligned. Please make sure to include the entire figure.
Lines 203, 206, and 210: insert a space after (or before) “sex” or “sexes”.
A comment on the Bland-Altman plots: For the newly-developed predicted equations it looks like REE is underestimated for the prediction equations at low metabolic rates and overestimated at high metabolic rates. Ideally you want to see the data points in the Bland-Altman plot evenly distributed above and below the zero line for all mean values. They tend to be below the zero line at low REEs and above the zero line at high REEs. I suggest adding a comment on this limitation of the new prediction equations in the discussion section.
Author Response
Thank you very much for taking the time to review this manuscript. Please find our responses below.
- Comment 1: Line 64: insert a space between the words “patients” and “and”.
- Response 1: Yes, we have inserted a space between the words, “patients” and “and”.
- Comment 2: Line 76: Delete the “g” at the start of this line.
- Response 2: Sure, we have deleted “g” accordingly.
- Comment 3: Line 96: For VO2 and VCO2 you have superscripted the “2”. This should be subscripted instead. Please correct this throughout.
- Response 3: Thank you for pointing this out, we have subscripted the VO2 and VCO2 in Page 3, Linee 115.
- Comment 4: Figure 1: This flow chart now runs off the page and is mis-aligned. Please make sure to include the entire figure,
- Response 4: Thank you for pointing this out, we have amended alignment of Figure 1.
- Comment 5: Lines 203, 206, and 210: insert a space after (or before) “sex” or “sexes”.
- Response 5: Yes, we have amended these accordingly.
- Comment 6: A comment on the Bland-Altman plots: For the newly-developed predicted equations it looks like REE is underestimated for the prediction equations at low metabolic rates and overestimated at high metabolic rates. Ideally you want to see the data points in the Bland-Altman plot evenly distributed above and below the zero line for all mean values. They tend to be below the zero line at low REEs and above the zero line at high REEs. I suggest adding a comment on this limitation of the new prediction equations in the discussion section.
- Response 6: Agree, we have included more information to Page 12, line 442-448 to explain this observation.